



# Aerosol characteristics and polarimetric signatures for a deep convective storm over north-western part of Europe – modeling and observations

Prabhakar Shrestha[1], Jana Mendrok[2], and Dominik Brunner[3]

[1]Institute of Geosciences, Meteorology Department, Bonn University, Bonn, Germany
[2]Deutscher Wetterdienst, Offenbach, Germany
[3]Empa, Swiss Federal Laboratories for Materials Science and Technology, Dübendorf, Switzerland

**Correspondence:** Prabhakar Shrestha (pshrestha@uni-bonn.de)

**Abstract.** The Terrestrial Systems Modeling Platform (TSMP) was extended with a chemical transport model and polarimetric radar forward operator to enable detailed studies of aerosol-cloud-precipitation interactions. The model was used at km scale (convection permitting) resolution to simulate a deep convective storm event over Germany, which produced large hail, high precipitation and severe damaging winds. The ensemble model simulation was in general able to capture the storm structure,

its evolution and spatial pattern of accumulated precipitation - however, the model was found to underestimate regions of high accumulated precipitation (> 35 mm) and convective area fraction in the early period of the storm. While the model tends to simulate too high reflectivity in the downdraft region of the storm above the melting layer (mostly contributed by graupel), the model also simulates very weak polarimetric signatures in this region, compared to the radar observations. The findings of the study remained almost unchanged when using much narrow cloud drop size distribution (CDSD) acknowledging the missing

feedback between aerosol physical and chemical properties and CDSD shape parameters.

The km scale simulation showed that the strong updraft in the convective core produces "aerosol tower" like features, increasing the aerosol number concentrations and hence increasing the cloud droplet number concentration and reducing the mean cloud drop size. This could be also a source of discrepancy between the simulated polarimetric features like differential reflectivity ($Z_{DR}$) and specific differential phase ($K_{DP}$) columns along the vicinity of the convective core compared to the X-

band radar observations. Besides, the evaluation of simulated trace gases and aerosols were encouraging, however a low bias was observed for aerosol optical depth (AOD), which could be partly linked to an underestimation of dust mass in the forcing data associated with a Saharan dust event.

This study illustrates the importance and the additional complexity associated with the inclusion of chemistry transport model when studying aerosol-cloud-precipitation interactions. But, along with polarimetric radar data for model evaluation,

it allows to identify and better constrain the traditional 2-moment bulk cloud microphysical schemes used in the numerical weather prediction models for weather and climate.



# 1 Introduction

The effect of aerosol on clouds and precipitation through microphysical and radiative processes, remains a major source of uncertainty in weather and climate prediction (Tao et al., 2012; Rosenfeld et al., 2014; Fan et al., 2016). In particular, improved

understanding of the microphysical pathways of how aerosol affects cloud evolution (e.g. Rosenfeld et al., 2008; Koren et al., 2008; Yuan et al., 2011; Storer et al., 2014; Jiang et al., 2018; Igel and van den Heever, 2021) and precipitation (e.g. Rosenfeld, 2000; Stevens and Feingold, 2009; Shrestha and Barros, 2010; Li et al., 2011; Guo et al., 2016, 2018) is important for better prediction of extreme events. Many sensitivity studies using numerical models with various degrees of sophistication have been conducted to better understand these microphysical pathways with idealized/semi-idealized (e.g. Khain et al., 2005; Tao

et al., 2007; Storer et al., 2010; Lebo and Seinfeld, 2011) or real data simulations (e.g. Noppel et al., 2010; Seifert et al., 2012; Morrison, 2012; Fan et al., 2013; Barros et al., 2018; Fan et al., 2018; Iguchi et al., 2020; Zhang et al., 2021; Trömel et al., 2021). While few of the above sensitivity studies have evaluated the model using radar reflectivity, polarimetric radar data which provides valuable information on cloud microphysical processes have not been fully exploited yet. In most of the numerical modeling studies, the aerosol physical and chemical properties have been held constant and a large-scale perturbation

of aerosol concentrations has been used for sensitivity studies. However, the classical assumptions made for "continental" or "marine" aerosols in the models do not reflect the actual local aerosol type, concentration and its vertical profile or temporal evolution for any particular region on the globe. In fact, the meteorological settings, land cover, land use and emissions strongly control the regional spectra of aerosol physical and chemical properties (e.g. Putaud et al., 2010; Martin et al., 2010; Shrestha et al., 2013). More recently, numerical modeling studies with a realistic aerosol distribution obtained by either downscaling

region-specific aerosol profiles from a global aerosol model or using a meteorological model online coupled to a chemistry transport model are emerging (e.g. Rieger et al., 2014; Iguchi et al., 2020; Zhang et al., 2021). However, these studies have not fully exploited the potential of evaluating the model simulations against polarimetric radar observations. In this study, we use an online coupled meteorology-chemistry model (Baklanov et al., 2014), the Terrestrial Systems Modeling Platform (TSMP; Shrestha et al., 2014; Gasper et al., 2014) with Aerosols and Reactive Trace gases (ART; Vogel et al., 2009) module for an

ensemble simulation of a summertime deep convective storm over Germany at km-scale (convection-permitting) resolution. The main goal of the study is to 1) extend the TSMP with a chemistry transport model and polarimetric radar forward operator to enable detailed studies of aerosol-cloud-precipitation interactions and their evaluation against polarimetric radar observations, and 2) to demonstrate these new capabilities for a case study of a deep convective storm over Germany.

The manuscript is arranged as follows: Sect. 2 describes the observation data, model, forward operator (FO) and model

setup used for the study. The first model evaluation of trace gases and aerosols with satellite and ground based observations is presented in Sect. 3. The modeled aerosol physical and chemical characteristics during the storm event are presented in Sect. 4. The evaluations of modeled cloud microphysical processes and precipitation using polarimetric radar data are presented in Sect. 5. A detailed analysis of polarimetric features and aerosol characteristics is presented in Sect. 6. Finally, discussion and conclusions are presented in Sect. 7 and 8, respectively.





## 2   Data and Methods

We study a summertime hail-bearing deep convective storm over north-western Germany. The northeastward propagating storm
was associated with the presence of pre-frontal convergence zones developed over this region on 5 July. Scattered storms were
prevalent throughout the day, with an isolated deep convective storm passing directly over Bonn, from 1500 to 1600 UTC
on 5 July 2015. Based on observations reported by the European Severe Weather Database (ESWD), large hail (2 - 5 cm in
diameter) was observed over the Bonn region, including damaging lightnings further north, and heavy precipitation with severe
wind (further north-east). A detailed discussion can also be found in (Shrestha et al., 2021).

We use the Bonn Radar domain (Shrestha, 2021) as a numerical modeling domain for this study. The study region encom-
passes the north-western part of Germany bordering with the Netherlands, Luxemburg, Belgium and France. The region is
characterized by multiple hills of the Rhine Massif with heights ranging from 600 to 800 m, and land cover including forest,
agricultural land, and urban/rural area. The region also comprises extensive emissions by point (e.g., oil refineries, other in-
dustries) and area sources (e.g. extensive urban and rural areas, road transport, extensive agriculture) (Kulmala et al., 2011;
Kuenen et al., 2014), which makes the region especially suited for this study. Additionally, due to the availability of the twin
polarimetric X-Band research radars in Bonn (BoxPol) and Jülich (JuxPol) and overlapping measurements from four polari-
metric C-Band radars of the German Weather Service (Deutscher Wetterdienst, DWD), along with the presence of the Jülich
Observatory for Cloud Evolution (JOYCE; Löhnert et al., 2015), the region probably represents the best radar-monitored area
in Germany.

Here, we extend TSMP with the ART module and use a forward operator to transform the model outputs into radar space for
evaluation with polarimetric radar observations from X-band radars. Available satellite observations and in-situ observations
are also used to evaluate the simulated trace gases and aerosols. A more detailed discussions about the observation data, model,
forward operator and the model setup are presented below.

### 2.1   Polarimetric Radar Observations

The attenuation-corrected polarimetric radar measurements from the twin research X-band Doppler radars located in Bonn and
Jülich (BoxPol and JuXPol; Diederich et al., 2015a, b) are used to investigate the microphysical characteristics of the deep
convective storm. The polarimetric radar measurements provide valuable information about horizontal reflectivity ($Z_H$), dif-
ferential reflectivity ($Z_{DR}$), specific differential phase ($K_{DP}$), and cross-correlation coefficient($\rho_{hv}$), which depend on hydrom-
eteor shape, orientation, density and phase composition, and thus enable a detailed evaluation of the modeled microphysical
and macrophysical processes. The radars are operating at a frequency of 9.3 GHz with a radial resolution of 100-150 m and a
scan period of 5 minutes. Both X-band Doppler radars produce volume scans at different elevations, mostly between 0.5 °and
30 °. These volume scans are also used to interpolate the polarimetric radar data from the native polar coordinates to Cartesian
coordinates at 500 m horizontal and vertical resolution using a Cressman analysis with a radius of influence of 2 km in the
horizontal and 1 km in the vertical. A threshold of 0.8 in $\rho_{hv}$ was imposed in the gridded data to ensure that clutter is filtered
out without removing useful meteorological information.





The polarimetric variables $Z_H$ and $Z_{DR}$ are potentially affected by radar miscalibration, partial beam blockage and (differential) attenuation, especially at smaller wavelengths (C band and X band), and their correction especially in deep convective,

hail-bearing cells gives rise to additional uncertainties (e.g. Snyder et al., 2010; Shrestha et al., 2021). Although $K_{DP}$ estimates are not affected by miscalibration and attenuation, they can be substantially affected by the uncertainty in the quantification of the backscatter differential phase ($\delta$), which is particularly important when hydrometeor sizes are in the range of, or larger than, the radar wavelength (Trömel et al., 2013). A more detailed discussion about the calibration, clutter filtering and attenuation correction of the polarimetric radar data can be found in Shrestha et al. (2021). It is important to note that errors in

estimates of polarimetric radar variables might arise due to the assumptions made in the attenuation correction algorithm and due to uncertainties in the contribution of backscatter differential phase to the total differential phase shift. We acknowledge these limitations in the study, and concentrate more on patterns and not so much on the actual magnitudes of the polarimetric moments. For precipitation, the RADOLAN (Radar Online Adjustment; Ramsauer et al., 2018; Kreklow et al., 2020) data from the German national meteorological service (DWD, Deutscher Wetterdienst) is also used for model evaluation. RADOLAN

is a gauge-adjusted precipitation product based on DWD's C-band weather radars available at hourly frequency in a spatial resolution of 1 km.

## 2.2  Trace Gases and Aerosols

The Ozone Monitoring Instrument aboard Aura satellite provides valuable observations to better understand the chemistry and dynamics of Earth's atmosphere (e.g., $O_3$, $NO_2$, $SO_2$, HCHO etc.). In this study, we make use of OMI $NO_2$ v4.0 data (Krotkov

et al., 2019; Lamsal et al., 2021) to evaluate the spatial pattern of modeled $NO_2$ vertical tropospheric columns (VTC). $NO_2$ is a key anthropogenic air pollutant and precursor of aerosols. The OMI estimates of VTC $NO_2$ are filtered for data points with VcdfQualityFlags=0 and CloudRadianceFraction < 0.5 (clear sky data). Similarly, the Moderate Resolution Spectroradiometer (MODIS) 3 km aerosol product (MOD04_L3; Levy and Hsu, 2015) is used to evaluate the simulated spatial pattern of aerosol optical depth (AOD) at 550 nm. In addition, AOD Level 2.0 (version 3) ground-based measurements from two Aerosol Robotic

Network (AERONET; Holben et al., 1998; Giles et al., 2019) stations over the domain are also used to evaluate the modeled AOD. These measurments have a better accuracy than MODIS but are only available at few locations.

## 2.3  Model

TSMP-ART v1.0 used in this study consists of the atmospheric model COSMO v5.1 (Consortium for Small-Scale Modeling; Steppeler et al., 2003; Baldauf et al., 2011) interfaced with ART v3.1 (Vogel et al., 2009), the land surface model CLM v3.5

(Community Land Model: Oleson et al., 2008), and 3D distributed groundwater model ParFlow v3.1 (Ashby and Falgout, 1996; Jones and Woodward, 2001; Kollet and Maxwell, 2006; Maxwell, 2013). The three component models are coupled using the OASIS-3 MCT coupler (Craig et al., 2017). COSMO-ART allows a comprehensive simulation of two-way interaction between full gas-phase chemistry and aerosol dynamics with atmospheric processes (e.g. aerosol direct and indirect effects; washout of aerosols). Since ART v3.1 is already available as a module for the COSMO v5.1 model (which can be turned on with pre-

processor flags), no extensive-additional work was required to include the ART module in TSMP. As such, TSMP software





was recently updated to include the ART v3.1 module with an extended version of the two-moment bulk microphysics scheme (Seifert and Beheng, 2006) including hail class (Blahak, 2008) (henceforth, SB2M). SB2M predicts the mass densities and number densities of cloud droplets, rain, cloud ice, snow, graupel and hail, which are the zeroth and first moments of the particle mass distribution (PMD) that is assumed to follow a modified gamma distribution (MGD)

$$f(x) = N_0 x^\mu \exp(-\lambda x^\nu) \qquad (1)$$

with $x$ being the particle mass and parameters $\mu$ and $\nu$ determining the shape of the distribution. The specific hydrometeor mass $q$ and specific number $n$ can be derived by $q = Q/\rho$ and $n = N/\rho$ with $\rho$ being the total density (air, vapor and hydrometeors).

The size-mass and velocity-mass relations of different hydrometeors are parameterized by power laws

$$D = a_g x^{b_g} \qquad (2)$$
$$v_T = a_v x^{b_v} \qquad (3)$$

with (maximum) particle diameter $D$, terminal fall velocity $v_T$ and parameters $a_g$, $b_g$, $a_v$ and $b_v$.

The shape parameters $\mu$ and $\nu$ of the MGD remain constant for each hydrometeor class and $N_0$ and $\lambda$ can be diagnosed from the two prognostic moments. Further, to mitigate unphysical effects on the mean spectral particle mass $\overline{x} = q/n$ coming from the separate advection and sedimentation of $q$ and $n$, a minimum and maximum allowable mass limit is imposed for $\overline{x}$ ($x_{min}$ and $x_{max}$) at relevant places during the model time stepping. This is done by clipping $n$ so that $\overline{x}$ stays within $[x_{min}, x_{max}]$. For reference, all fixed parameters which were used in this study are summarized in Table 1.

When the SB2M is coupled with the ART module, the cloud nucleation parameterization is based on the works of Fountoukis and Nenes (2005), Barahona and Nenes (2007), Kumar et al. (2009) and Barahona et al. (2010). Similarly, the ice nucleation parameterization is based on Barahona and Nenes (2009). A more detailed discussion about the implementation of the above nucleation parameterizations in ART is available from Bangert et al. (2012). Also, the parameterizations for the direct aerosol effect on radiation and washout of aerosols by precipitation was turned on for the simulations. These formulations are all based on the prognostic aerosol population with 12 overlapping modes simulated in ART. Each mode is approximated by a lognormal distribution with uniform chemical composition across size. The 12 modes consist of: nucleation and accumulation mode for pure and mixed aerosol particles (sulphate, ammonium, nitrate, organic compounds, water, and soot); small, medium and large particles for dust and sea-salt; soot particles; and coarse particles (not used for the nucleation parameterization). These aerosol modes are coupled with gas-phase chemistry and strongly influenced by the atmospheric boundary layer evolution, advection and anthropogenic emissions of gases and particles. An additional overview about the individual aerosol modes, chemical composition and cloud interactions processes along with the aerosol dynamical processes can be found in (Bangert et al., 2012).

For input of emission inventories, the online emission module developed earlier by Jähn et al. (2020) is used. This module makes use of pre-processed inventory data projected onto the model grid along with temporal and vertical scaling profiles for individual emission categories. A more detailed discussion about the pre-processing of emission inventories is presented in section 2.5.





## 2.4 Forward Operator

EMVORADO, the Efficient Modular VOlume RADar Operator, (Zeng et al., 2016) is COSMO's native radar forward operator. FO requires consistency with the model, particularly regarding hydrometeor microphysics, i.e., size distributions as well as mass-size and velocity-size relations. For the online version run simultaneously with COSMO, this is ensured completely through variables shared between the modules. For offline version run, this consistency is maintained manually. Here, we make use of the offline version, though, which is more flexible and allows to re-run the FO with varied in-FO assumptions for, e.g.,
sensitivity analyses.

For offline run, EMVORADO requires as input the atmospheric fields of mass and (for SB2M) number concentrations of the six hydrometeor classes (cloud liquid, rain, cloud ice, snow, graupel and hail), of temperature and of the three wind components. Other parameters the affect forward modelled polarimetric radar observables are insufficiently constrained by the COSMO model and assumptions need to be made within the FO. This regards, e.g., the phase partitioning of hydrometeors
during melting, the shape and orientation of particles, and the heterogeneous microstructure of frozen hydrometeors.

Like essentially all bulk scheme models, SB2M does not provide a prognostic melt fraction and hydrometeors are either (completely) frozen or liquid. All meltwater is assumed to be shed instantaneously and transferred into the rain hydrometeor class, hence no mixed-phase hydrometeors are predicted. Liquid water and ice exhibit significantly different dielectric properties in the radar frequency region, which leads to strong changes in the reflectivities where a phase change takes place. The
melting layer is hence appearing very prominently in radar observations, particularly in stratiform situations, as layer of enhanced reflectivity known as the radar bright band. In order to be able to simulate such features, the forward operator needs to employ a melting scheme that predicts the occurrence of mixed-phase, "wet" frozen hydrometeors based on the single-phase model hydrometeors. EMVORADO employs a melting scheme that assumes a certain fraction of the frozen hydrometeor mass to be liquid (in contrast to, e.g., Wolfensberger and Berne (2018); Jung et al. (2008), which redistribute a part of the rainwater
back into the frozen hydrometeor classes, i.e. "unshed" some rainwater). EMVORADO models the liquid water fraction dependent on the size of the hydrometeors (considering that small particles melt faster than large ones) and the ambient temperature $T$ (Blahak, 2016). Wet hydrometeors start to occur when $T$ exceeds a threshold $T_{\text{meltbegin}}$ and are assumed to be completely melted when $T_{\text{max}}$ is reached, where $T_{\text{max}}$ by default is determined dynamically from the model hydrometeor field and $T$ in the vertical column. Setting $T_{\text{meltbegin}}$ accordingly, this allows for wet frozen hydrometeors at sub-zero temperatures, covering the
case of upward transport of liquid water and wet hydrometeors that do not (re-)freeze instantaneously in convective updrafts. Through the temperature dependence parameters, which are specific to each frozen hydrometeor class, the melting scheme can be adjusted by the user. Unless noted otherwise, in this study we apply EMVORADO's default melting scheme parameters (see Table 2).

Shape and orientation of the hydrometeors significantly affect the polarimetric radar parameters, but are entirely uncon-
strained by the COSMO model. Here we make use of the polarimetric mode of EMVORADO, which so far applies the T-Matrix scattering method for one- (Mishchenko, 2000) or two-layered (Ryzhkov et al., 2011) spheroidal particles. All hydrometeors are assumed as oblate spheroids (except for liquid cloud particles that are modelled as spheres using Rayleigh scattering) with





hydrometeor class specific, size and melt fraction dependent parameterisations of shape and orientation of the hydrometeors as given by Ryzhkov et al. (2011). Effect of orientation distributions are considered using the angular moments approach outlined in Ryzhkov (2001); Ryzhkov et al. (2011, 2013).

In order to allow fast calculation of the radar observables, lookup tables of bulk scattering properties are pre-calculated tabulating basic (additive) quantities per hydrometeor class over bulk (mean) mass, temperature and melting $T_{\mathrm{max}}$. These are then added up over the six hydrometeor classes and converted into the polarimetric radar observables. Beside the reflectivity factor in horizontal polarisation $Z_{\mathrm{H}}$, in this study we focus on the differential reflectivity $Z_{\mathrm{DR}}$, the co-polar cross-correlation coefficient $\rho_{\mathrm{HV}}$ and the specific differential phase $K_{\mathrm{DP}}$. In short, $Z_{\mathrm{DR}}$ is the difference of the (log- or dBZ-space) reflectivities in horizontal and vertical polarization, $\rho_{\mathrm{HV}}$ the correlation between reflectivities in horizontal and vertical polarization within the measurement volume and $K_{\mathrm{DP}}$ the phase difference between the horizontal and vertical polarized wave returns. A more comprehensive description can be found, e.g., in Kumjian (2013).

EMVORADO is capable of simulating the sensing process, including scanning, beam tracing, beam blockage, beam pattern, attenuation. This allows to directly simulate observation equivalents like 3D volume scans. However, here we make use of the radar parameters calculated on the model grid, i.e., neglecting any sensing effects.

## 2.5 Model Setup

The simulation is set up for an approximately 340 km x 340 km wide Bonn Radar domain at km scale resolution for the period 4 to 5 July 2015. For the initial and lateral boundary condition (IC/BC) of the atmospheric states in COSMO, data from the COSMO-DE Ensemble Prediction System (EPS; Gebhardt et al., 2011; Peralta et al., 2012) are used. The EPS data represents uncertainties in model physics and lateral boundary conditions by combining five model physics perturbations with four global models. An earlier study by Shrestha et al. (2021) showed that the statistics of the EPS are always stratified according to the four global models; i.e. the five members having the same global model are more similar to each other. In this study, we therefore only employ those 5 ensemble members that are based on the same global model of DWD (GME; Majewski et al., 2002). The ensemble simulation is used to reduce uncertainty associated with meteorological forcings. The initial soil-vegetation states for CLM and ParFlow are obtained from spinups using offline hydrological model runs over the same domain (Shrestha, 2021). For initial and lateral boundary condition of trace gases and aerosols, we use the 6-hourly data from Model for Ozone and Related chemical Tracers, version 4 (MOZART-4; Emmons et al., 2010). The MOZART-4 data is available at a resolution of 1.9 °x 2.5 °with 56 levels (https://www.acom.ucar.edu/wrf-chem/mozart.shtml). The COSMO model Processing Chain version 2.2 (available from https://github.com/C2SM/processing-chain) was used for pre-processing of the MOZART-4 data into ART variable states. This python script maps the gases and aerosols (mass concentrations) from MOZART-4 to ART state variables. For initialization of the number concentration of each aerosol mode, the default density and initial mode diameters in the ART module are used. Further, we also assume that the aerosol has been in the atmosphere for a long time, where it could coagulate and mix, so 0.1 and 99.9 % of the fine mode aerosols are assigned to mixed nucleation and accumulation mode, assuming a median diameter of these modes of 50 and 150 nm respectively. The mapping from MOZART-4 to ART aerosol classes and the assumptions regarding median diameters are an additional source of uncertainty in the initialization of aerosols in the model.





The Copernicus Atmosphere Monitoring Service – Regional Inventory v4.2 (CAMS-REG v4.2; Kuenen et al., 2022) was used to prescribe the spatiotemporal emissions for the study. CAMS-REG v4.2 is a state-of-the-art gridded anthropogenic emission inventory developed for the European domain at a $0.1\,°\times0.05\,°$ grid resolution, with a temporal coverage of 18-years

(2000–2017). This emission inventory was pre-processed using the Python package "emiproc" (Jähn et al., 2020), available publicly through the C2SM GitHub organization (https://github.com/C2SM-RCM/cosmo-emission-processing) for COSMO-ART variable states. First, the emission inventory data is projected onto the model grid and then the temporal and vertical scaling profiles for individual emission categories are estimated. These inputs are then read during the model initialization and the temporal and vertical emission profiles per category are applied online during the model run. In addition, the land cover

data from Global Land Cover Map for 2009 (GlobCov 2009; Arino et al., 2012) is used for the biogenic volatile organic carbon (VOC) emissions. Further, there is no emission of dust inside the model, and dust only comes from the MOZART-4 boundary conditions.

The ensemble simulation starts on 4 July 2015 0600 UTC and the model is integrated for 42 hours. In all runs, a coupling frequency of 90 s is used between the atmospheric and hydrological components, which have a time step of 10 s and 90 s,

respectively. The model is integrated over diurnal scale and the output is generated at 5 min intervals and hourly intervals for evaluation with polarimetric radar data (only for a 3 hour period) and aerosol measurements respectively.

## 3 Evaluation of simulated trace gases and aerosols

First, the modeled trace gases and aerosols are evaluated with satellite and ground based observations. For comparison, the model data was also cloud screened. A threshold of $20 g/m^2$ was used for the total column integrated liquid and ice condensate

for the cloud screening.

The vertical tropospheric column (VTC) is used to compare simulated $NO_2$ with satellite estimates from OMI. The VTC is an integral measure of the tracer from the surface to the tropopause. While it can be readily estimated from the model, the satellite estimates are dependent on the assumed vertical profiles of $NO_2$ in their algorithms. We acknowledge this uncertainty in the satellite estimates and the corresponding limitations of a direct comparison with the model data. However, it has to be

stressed that this comparison still provides a first order evaluation of modeled trace gases, which is important for this study. Both the satellite and the modeled VTC for $NO_2$ exhibit similar patterns, with relatively higher magnitudes over the northern-western low lands, and lower magnitudes over the Rhine massif around 4 July 2015 1200 UTC (Fig. 1). However, the model exhibits relatively higher magnitude of $NO_2$ over the foothills of the Rhine massif near the emission sources (mostly from the mining regions and industry north-west of Bonn), which is not captured in the satellite retrievals.

The modeled aerosol optical depth (AOD) is also compared with satellite retrievals from MODIS. In comparison with the MODIS data on 4 July 2015, the model tends to simulate relatively low AOD (0.1 0.3) over most of the domain (Fig. 2a-b). The MODIS data also show low AOD (0.1 0.3) over large parts of the domain but with pockets of high AOD scattered over the northern parts, which is not captured by the model. This bias in the modeled AOD can also be observed when comparing the modeled AOD with available AERONET station data over the region. The model generally tends to underestimate the AOD as





255 estimated by the in-situ measurements (when available). This is more prominent for the MAINZ station (Fig. 2d). However, within the spread of the ensemble members, the model also tends to the capture measured AOD over some period of times at FJZ-JOYCE station (Fig. 2c).

  In general, the above model evaluation with satellite data and ground based measurements do build some confidence over the modeled aerosol and gaseous species.

## 4 Aerosol Characterization

260 The TSMP-ART simulated evolution of aerosol physical and chemical properties during the convective storm event are summarized in Fig. 3 and Fig. 4 respectively. Two different time periods are chosen as the storm propagates toward north-west, with strong updrafts in the synthetic sampling location at 1500 UTC. The aerosol number concentrations of different modes ($N_x$) exhibit a strong variability in space and time. Fig. 3a shows the spatial pattern of number concentrations of the sum of

265 nucleation and accumulation mode for both pure and mixed aerosols ($N_{na}$) at 2 km height on 2015 July 5 1400 UTC. At this time, the sampling location exhibits relatively low $N_{na}$ compared to the western part, which has an extended patch of high $N_{na}$ extending from east to west. Over the next hour, this patch appears to be advected north-west owing to the dominant north-westerly wind direction (Fig. 4a). However, at the same time the spatial propagation of convective updrafts also plays a crucial role in lifting of aerosols to 2 km altitude. The evolution of the spatial pattern thus appears to determined by a combination

270 of horizontal advection and vertical updraft, the latter additionally depending on the co-location with local emissions. Fig. 3b shows the average aerosol size distribution for different modes and PM2.5 (particulate matter with size $< 2.5 \mu m$) chemical composition for a 9x9 grid cells box encompassing BoxPol at center. At 2 km height, the dust particles dominate the aerosol mass while soluble components make up only about 26 %. As expected, $N_{na}$ is highest near the surface and generally decays with height. The magnitude of $N_{na}$ is around $180 \, cm^{-3}$ at 2 km level. Also, a rightward shift in the aerosol size distribution

275 of nucleation/accumulation mode can be observed associated with fresh aerosols near the surface to more aged aerosols in upper layers with a larger mode around 300 nm. The soot particles exhibit a multi-modal distribution with larger mode around 200 nm, while the dust particles exhibit a larger mode around 2000 nm. Fig. 3c shows the meridional cross-section of aerosol number concentration ($N_x$) for combination of different aerosol modes. As observed in the aerosol size distribution (Fig. 3b), $N_{na}$ and $N_{soot}$ exhibit higher concentration below 3 km height. Localized high values of $N_{na}$ along the cross-section are

280 associated with local emissions. The dust aerosols exhibit a more horizontally homogeneous profile with a peak around 4 km, probably associated with a Saharan dust event. The multi-model forecast of dust from World Meteorological Organization (WMO) Barcelona Dust Regional Center (https://dust.aemet.es/products/daily-dust-products) indicates the presence of Saharan dust for this particular event. The PM2.5 concentration also shows peaks near the surface and near the melting layer but associated with the $N_{na}$ and $N_{dust}$ respectively.

285 Fig. 3d shows the area average vertical profile of aerosol number concentration for different modes. The profiles are shown for the same 9x9 grid cells for 5 ensemble members. In general, all ensemble members exhibit similar profiles for this time period. Importantly, the aerosols exhibit a strong diurnal cycle owing to emissions, ABL evolution, updraft and



advection (Fig. 3e). At 2 km height, it generally peaks during the day and decays during the night (here only shown for nucleation/accumulation mode), except for periods with persistent convection or advection of aerosols, as observed for example

as a sudden increase in aerosol number concentration from 1400 to 1500 UTC on 5 July (blue to red line). The situation at 1500 UTC, when the aerosol distribution is strongly influenced by the deep convective event, is illustrated in Fig. 4. Due to the strong updraft associated with the convective storm, the aerosol size distribution of nucleation/accumulation mode has become much broader (especially at 2-4 km height) as compared to the situation at 1400 UTC. At the same time, the aerosol number concentration has increased and the chemical composition (Fig. 4b) has changed significantly. The aerosol solubility and

PM2.5 concentrations at 2 km height have increased rapidly from 26 % to 46 % and 6.77 $\mu g m^{-3}$ to 9.49 $\mu g m^{-3}$, respectively. The simulated strong updraft over the sampling location also appears to generate localized "aerosol towers" reaching up to 15 km height (Fig. 4c). This increases the aerosol number concentration for all modes rapidly at higher altitudes (see Fig. 4d). The spread in the ensemble members is associated with the variability in the location and magnitude of the simulated updraft associated with the convective storm.

## 300  5   Evaluation with polarimetric radar data

First, the modeled daily accumulated precipitation (5 July 2015) is evaluated with estimates from RADOLAN. Fig. 5a-b shows the spatial pattern of ensemble averaged and RADOLAN accumulated precipitation. In general, the model is able to capture the spatial pattern of the observed precipitation. However, the model underestimates the high precipitation in the north-east part of the Bonn radar domain. This underestimation is also seen in the frequency distribution of the simulated and observed

accumulated precipitation (Fig. 5c). While the domain average precipitation is similar to the RADOLAN data, all ensemble members tend to underestimate regions with high accumulated precipitation (>35 mm). But, all ensemble members tend to slightly overestimate medium accumulated precipitation (10 to 30 mm).

The underestimation of high accumulated precipitation indicates that the model underestimates the high precipitation amounts associated with the core of the convective storm. This is also well seen in the time-series of the convective area fraction (CAF;

Fig. 6), which is estimated as the ratio of storm area at 2 km above ground level (a.g.l. hereafter) with $Z_H$ >40 dBZ to the total area with $Z_H > 0$ dBZ. The masked storm area is generated using a storm tracking algorithm, which uses edge detection and overlapping areas between consecutive snapshots to track the storm.

Observations from JuXPol and BoxPol exhibit high values of CAF in the early period of the storm (1400 to 1530 UTC), which is underestimated by all ensemble members. The ensemble members exhibit a similar pattern with increasing CAF

after 1530 UTC, when the simulated CAF matches more closely the observed CAF. However, such direct comparisons are always challenging due to mismatches in simulated and observed storm evolution in space and time, so we also conducted a qualitative exploratory analysis (using synthetic polarimetric variables at lower altitude (1 km a.g.l.), near the melting level (4 km a.g.l.) and at higher altitude (7 km a.g.l.) to find simulated convective storm structures closer in time and space to the radar observations. Based on this analysis, we compare the polarimetric signatures of the storm between one of the ensemble

members (solid line; Fig. 6) and the BoxPol observations at 1530 UTC.





Fig. 7a) shows the plan position indicator (PPI) of polarimetric variables at 8.2 ° elevation from BoXPol measurements. Near the melting level, the storm is characterized by high reflectivity (>50 dBZ) and differential reflectivity (>2 dB). At upper levels (beyond the convective core), the storm exhibits reflectivity in the range of 15 to 25 dBZ. The inflow region of the storm lies in the south-east corner which has relatively lower $\rho_{HV}$ but high $Z_H$ and $Z_{DR}$. The storm also exhibits an arc like feature of high

$Z_{DR}$ along the eastern edge. Fig. 7b) shows the vertical cross-section of the same polarimetric variables based on the gridded radar data along a north-south transect through the storm center. The convective core extends from -20 to 5 km, exhibiting high reflectivity (>50 dBZ) from surface up to 6 km height. A well defined $Z_{DR}$ column (>2 dB) anchored to the surface and extending up to 6 km height is also visible along the cross-section. $Z_{DR}$ columns are distinct polarimetric signatures often found along the vicinity of the strong convective updraft core (Kumjian and Ryzhkov, 2008; Kumjian et al., 2014; Snyder

et al., 2017b). $K_{DP}$ columns (Ryzhkov and Zrnic, 2019; Snyder et al., 2017b) are also clearly distinguishable and co-located with $Z_{DR}$ columns with slight inward offsets. High $Z_{DR}$ and $K_{DP}$ above the melting layer often indicate the presence of frozen raindrops, water coated hail and large size supercooled raindrops (Kumjian et al., 2014; Ryzhkov and Zrnic, 2019). Below the melting layer in the convective region, $K_{DP}$ also has high magnitudes contributed by the melting of graupel/hail into raindrops. The high reflectivity values in the convective core with low $\rho_{HV}$ (<0.92) also indicates the dominance of hail signature.

Fig. 8a-b) shows the spatial pattern of the synthetic polarimetric variables for the storm at 1 km and 4 km a.g.l derived from the model simulation. Compared to the observations, the storm is already ahead of the BoxPol location, but exhibits a similar structure compared to the observations. At lower levels (1 km a.g.l.), the storm exhibits an elongated zone with $Z_H > 40$ dBZ which is also associated with relatively high $Z_{DR}$, $K_{DP}$ but relatively lower $\rho_{HV}$. Near the melting level, the extent of the region with $Z_H > 40$ dBZ is much wider, also partly associated with high values of $K_{DP}$. However, relatively high values of

$Z_{DR}$ and lower values of $\rho_{HV}$ are mostly constrained around the convective core. The meridional cross-section of the synthetic polarimetric variables show that the storm top extends up to 13 km height, with an overshooting top up to 15 km height (Fig. 8c). The convective core also exhibits reflectivity > 50 dBZ up to 10 km height, but is relatively narrow compared to the observation. A $Z_{DR}$ column like feature protruding on top of the melting layer and anchored to the ground is also simulated, however its magnitude is less than 0.8 dB. This is much weaker than the observed $Z_{DR}$ columns with a magnitude > 2 dB. Above the

melting layer, $Z_{DR}$ is generally weak (0 to 0.1 dB) with slightly higher values along the convective core. $K_{DP}$ also exhibits relatively high values in the convective core extending up to the storm top. $\rho_{HV}$ is also relatively lower in the convective region and below the melting layer. In general, there is lack of polarimetric signal above the melting layer in the downdraft region of the storm, similar to an earlier study by Shrestha et al. (2021). The low variability and high values of synthetic $\rho_{HV}$ can be attributed to the shortcomings in FO assumption of hdyrometeor shape and orientation Shrestha et al. (2021). The lack of

polarimetric signature in the downdraft region of the storm above the melting layer could be due to the deficiency in the FO to correctly model the scattering properties of snow and graupel which dominate this region, as discussed below.

The meridional cross-section of the modelled hydrometeors shows the presence of supercooled rain drops in the strong updraft region, where the modelled vertical velocity above 8 km reaches 40 m/s(Fig. 8d). The strong updraft also generates a warm anomaly above the melting layer (see the 0 °C isotherm), below which rain is mainly produced by melting of graupel

and hail. The melting of graupel and hail into raindrops produces the high $K_{DP}$ below the melting layer. For ice hydrometeors,





graupel dominates, with high density surrounding the convective core. Graupel is responsible for the high reflectivity in the downdraft region of the storm above the melting layer. Cloud ice is located mostly above 8 km height and contributes to the high $K_{DP}$ near the storm top. The self-collection of these ice particles leads to the formation of snow which extends further down to 6 km as it grows in size via aggregation. Hail mostly dominates in the strong updraft region of the storm with peaks

in mass density adjacent to the supercooled rain drops. It also contributes to the high $Z_{DR}$ values simulated in the convective region above the melting layer. The mean diameter of the supercooled raindrops are only around 0.1 to 0.3 mm, and the above observed $Z_{DR}$ column-like signature is produced by the presence of water above freezing level due to melting of hail only. The mean hail size ranges from 0.1 to 13 mm (e.g., around 6 km height). During this time, the hail is also reaching the ground starting from 1525 to 1540 UTC. In general, $Z_{DR}$ column usually appears 15-20 minutes before the hail reaches the ground

(Ilotoviz et al., 2018). So, we also additionally explore this polarimetric feature in detail at earlier times in the following sections.

While the above analysis already indicates some shortcomings in the synthetic polarimetric signatures, the uncertainty due to mismatches between space and time scales of synthetic and observed polarimetric variables also needs to be addressed by monitoring ensemble properties of the convective storm. So, additionally, the synthetic polarimetric variables from the

ensemble simulations are compared to the observations from 1445 to 1530 UTC (see Fig. 6) using contoured frequency altitude diagrams (CFADS; Yuter and Houze Jr, 1995)) using the same extents and bin widths.

Fig. 9a shows the CFADs of the polarimetric variables from BoXPol measurements. $Z_{H}$ exhibits a narrow distribution at upper levels, with peaks around 15 to 25 dBZ. The distribution gradually broadens from the mid levels to the ground, with peaks around 15 to 40 dBZ near the melting layer. $Z_{DR}$ has a narrow unimodal distriution above the melting layer with peak

around 0.14 dB. The distribution gradually broadens below the melting layer with peaks shifting to 0.62 dB near the lower levels. $K_{DP}$ also has a very narrow unimodal distribution with peak around 0.1 deg/km. The distribution does exhibit a weak broadening from 8 km towards the surface. $\rho_{hv}$ exhibits a broader distribution with peaks around 0.97 to 0.99 upto 10 km height. Above, the peak shifts towards 0.82 to 0.85.

Compared to the observations, the ensemble CFADs of synthetic $Z_{H}$ exhibit a relatively broader distribution with peaks

around 20 to 35 dBZ in the upper levels. The peak of the distribution gradually shifts rightwards (30 to 40 dBZ) near the melting layer. Below the melting layer, the peak of the distribution shifts leftwards (3 to 25 dBZ), which also explains the lower CAF from the ensemble members compared to observations during this period. Similar to observations, synthetic $Z_{DR}$ has a narrow distribution above the melting layer with peak around 0.11 dB.The distribution gradually broadens below the melting layer with additional peaks around 1 and 2 dB. Similar to observations, synthetic $K_{DP}$ has a very narrow unimodal distribution

with a peak around 0.12 deg/km. It also exhibits a weak broadening in the storm top region and near the melting layer. For synthetic $\rho_{hv}$, the ensemble model CFADs show a very weak variability with a peak around 0.99 and a slight broadening below the melting layer.





## 6 Polarimetric Feature and Aerosol Characteristics

The observations from X-band radar show the $Z_{DR}$ and $K_{DP}$ column as one of the distinct features of this storm. However, the
model is only able to simulate comparatively weak polarimetric features. These polarimetric features and aerosol characteristics
are therefore explored also for an earlier time at around 1500 UTC. This time was chosen because it is also 25 minutes ahead
of hail reaching the ground, and the availability of additional aerosol data (due to hourly output).

Fig. 10a,b) shows the horizontal plane of $Z_{DR}$ at 6 km height along with the vertical wind speed. A strong convective core
is visible with a width around 12 km and vertical speed exceeding 30 m/s. Also, enhanced $Z_{DR}$ is visible surrounding the inner
convective core. The forward flank downdraft and the rear flank downdraft is also visible. The meridional cross-section shows
the presence of warm temperature perturbation above the melting layer in the convective core, which is mainly responsible for
the melting of hail in the FO at relatively higher level, producing the ring-like $Z_{DR}$ feature around the convective core.

The simulated $Z_{DR}$ signal is mostly produced by the rain drops and hail particles (Fig. 10c,d). Rain-drops dominate the
convective core (above the melting layer) and downdraft region (below the melting layer) in terms of mass density. Hail mostly
peaks north-west of the strongest convective core, and also extends partly to the downdraft region above the melting layer.
Above the melting layer in the convective core, the mean raindrop size is only around 0.1 to 0.3 mm, while the south-west
region does exhibit grid-scale supercooled raindrops with size range 1-3 mm, but part of its polarimetric signal is also masked
by hail in the FO. The mean size of raindrops below the melting in the downdraft region is around 1 to 3 mm, which contributes
strongly to the $Z_{DR}$ signal besides the contribution from melting hail. The mean size of the hail particles is generally around 1
to 13 mm with peak values around 6 to 9 km.

The comparatively small mean size of the hail particles and raindrops in the convective core could be due to the very high
cloud drop number concentration simulated in the model (Fig. 10e,f). The cloud drop number concentration exhibits strong co-
variability with the simulated nucleation/accumulation mode aerosol number concentration ($N_{na}$). The strong updraft increases
the aerosol load in the convective core, which increases the aerosol number concentration and consequently the cloud drop
number concentration, which varies from 100 to 3000 $cm^{-3}$ leading to very small size of cloud drops ranging from 5 to
25 $\mu m$.

## 7 Discussion

In this study, we extended the state-of-the-art terrestrial systems modeling platform with a chemistry transport module and
a polarimetric forward operator. The model was then used to evaluate synthetic polarimetric signatures of a deep convective
storm event over Germany with observations from X-band radar to better understand aerosol-cloud-precipitation interaction.

The model was also evaluated with satellite and ground based observations of trace gases and aerosols. The spatial pattern of
$NO_2$ VTC was well captured by the model. This is consistent with earlier evaluation of COSMO-ART by Knote et al. (2011),
who also showed that the model was able to capture the spatial pattern and magnitudes compared to OMI estimates over
Europe. Their study also showed that COSMO-ART underestimated the summertime AOD over much of Europe, compared
to the estimates from MODIS, which is also consistent with the findings in this study. This indicates a possible model bias,





which could be attributed to missing aerosol mass at lateral boundaries as well as inaccuracies in simulated aerosols within the domain (Knote et al., 2011). Additionally, the WMO Barcelona Dust Regional Center multi-model forecast shows dust AOD of 0.1 to 0.2 for this event, however the model estimates of dust AOD are much lower around 0.04 to 0.06. This indicates that dust mass was possibly underestimated in the MOZART-4 data used in this study, which could be contributing the low bias of
the simulated AOD.

In contrast to studies using fixed (e.g. climatological) aerosol distributions and properties, accounting for the full life cycle of aerosols using the ART v3.1 module introduces a strong diurnal cycle of aerosol physical and chemical properties, which are further modulated by synoptic winds and local convection. The typical large mode of the aerosol is around 300 nm, which is consistent with the assumptions made in SB2M runs (without the inclusion of ART v3.1 module). However, the number
concentrations and chemical composition (hence solubility) of the aerosol exhibit strong variability in space and time. E.g., during the convective event, the aerosol concentration and solubility at 2 km height rapidly increased. But, it has to be noted that the model could also be overestimating number concentrations near the ground, as found in earlier model evaluation study by Knote et al. (2011) for many regions in Europe. The model simulation also shows a rapid increase in aerosol concentrations within the convective storm up to the overshooting cloud tops, generating "aerosol towers" with contrasting aerosol properties
within and outside the storm. But, the uncertainty in the parameterization of the in-cloud processing of aerosols could also contribute to uncertainty in the simulated aerosol properties within the storm (Knote and Brunner, 2013).

In terms of accumulated precipitation, the model is able to capture the spatial pattern but underestimates the observed high precipitation amounts (> 35 mm) for all ensemble members. This finding is similar to results from an earlier study using TSMP with prescribed continental cloud nuclei (CN), and default ice nuclei (IN) concentrations (Shrestha et al., 2021). Also,
similar to the finding in this study, the CAF is also underestimated in the early phase of the storm (1445 to 1530 UTC), compared to the observations. The underestimation of CAF could be associated with 1) reduction in collision and coalescence efficiency associated with small size of cloud droplets, 2) strong updrafts and high aerosol number concentrations, and 3) missing feedback between aerosol number concentrations and shape parameters governing cloud drop size distribution. The km-scale resolution of the current modeling study could be contributing to model induced circulation enhancing the updraft
speed (Poll et al., 2017; Poll et al.), while the high aerosol number concentrations in the convective core resulting from the strong updraft contributes to large number concentration of small cloud droplets.

In general, all ensemble members are able to capture the storm structure and evolution similar to the observations. However, the polarimetric signals above the metling layer is generally weak in the downdraft region as also observed in earlier study (Shrestha et al., 2021), and also has higher reflectivity range compared to the observation. This is well captured in the CFADs
compared to the observations. Above the melting layer, the model generally overestimates the horizontal reflectivity compared to the observation, which is primarily due to overproduction of graupel in the model. The predefined ice categories with fixed properties in bulk microphyiscs scheme (e.g. SB2M used here) do not allow the simulation of full growth process for rimed particles like graupel or hail. This could contribute to the model bias in reflectivity in the downdraft region above the melting layer. A recent study by (Milbrandt et al., 2021) have also shown that the 3-moment representation of ice hydrometeors with





the predicted particle properties (P3 scheme; Morrison and Milbrandt, 2015) improves the simulated reflectivity above the downdraft region for a hail-bearing storm.

In terms of observed polarimetric features, the synthetic polarimetric variables also exhibit $Z_{DR}$ column like feature (though of much weaker magnitude) along the updraft region as in the observations. This difference may be attributed to a too small size of supercooled rain drops, but it may also be associated with the deficiency in the simulated updraft structure, recirculation

of raindrops and treatment of slow freezing of raindrops (Kumjian et al., 2014; Snyder et al., 2015). Also, importantly, the $Z_{DR}$ signal contribution from water-coated hail owing to wet growth process is missing. The current FO only has parameterization for melting of hydrometeors, but the water-coated hail particles due to wet growth is not included. Further, the collision efficiency between frozen particle and supercooled droplets decreases with drop size, resulting in a weaker riming and hence producing smaller hail particles with lower fall velocity (Noppel et al., 2010). The study by (Noppel et al., 2010) using the

COSMO model with SB2M microphysics showed that the continental CN concentration led to a weaker hail storm, however, additional sensitivity study by varying the shape parameters for cloud droplets producing narrow distribution led to a different conclusion indicating that the missing feedback between the shape parameters of cloud droplets and CN concentrations. So, we also conducted an additional ensemble sensitivity study using a narrow cloud droplet size distribution (CDSD;see Fig. 11a). The parameters $\mu$ and $\nu$ determining the shape of the distribution was changed to 6 and 1 respectively from the default value

used in this study, and referred to as narrow CDSD. With the narrow CDSD runs, all ensemble members still underestimate the CAF. The domain average precipitation and the spread of the frequency distribution of the precipitation are similar to the default runs. Only the spatial location of high precipitation for the ensemble average in the north-eastern part of the domain is slightly shifted (Fig. 11b). Besides, the narrow CDSD does effect the CFADs of the storm in terms of polarimteric variables. At upper levels, the peaks of $Z_{H}$ shifts to higher magnitudes at 20 to 35 dbZ. And, the distribution gradually broadens and the peak

shifts rightward (30 to 40 dBZ) near the melting layer. Below the melting layer, the distribution shifts rightward as simulated before (with default CDSD), with peaks around 5 to 25 dBZ. CFADs of $Z_{DR}$ also exhibits multimodal distribution below the melting layer with additional peaks around 0.87 dB and 1.87 dB, which is slightly lower than the default CDSD runs. Above the melting layer, the peak of $Z_{DR}$ remains at 0.11 dB. The CFADs of the KDP and $\rho_{HV}$ also exhibit a similar peak around 0.11 deg/km and 0.99 respectively. But, in general, there is an increase in the the spread of all the polarimetric varaibles. This could

probably indicate the importance of the shape parameters of the hydrometeors to improve the simulated polarimetric signature of the storm.

## 8  Conclusions

While acknowledging the model biases and uncertainty in the simulated aerosol properties, the inclusion of prognostic aerosol is a way forward in better understanding the aerosol-cloud-precipitation interactions. During the convective storm event, the

model generates "aerosol tower" like features with contrasting physical and chemical properties compared to the background.

At diurnal scales, the model is able to capture the spatial pattern of the precipitation, however the comparison with the polarimetric observations indicate possible deviation in the ice hydrometeor partitioning above the melting layer (especially



in the downdraft region of the storm), size of supercooled rain drops and hail in the vicinity of the convective core and the mechanism of rain production below the melting layer - hence the particle shapes and concentration. Besides the shortcomings in the traditional 2-moment bulk scheme used in this study, the effect of model grid resolution and its impact on the structure of the storm updraft, and the effect of simulated high number of aerosol concentration which gets lifted in the convective core and hence the polarimetric signature in the vicinity of the convective core can also be not neglected.

Thus, future aerosol-cloud-precipitation interaction studies using models should make an effort to include prognostic aerosol models and evaluate the cloud microphysical processes using polarimetric radar data to identify and improve the cloud microphysical parameterization in the current NWP model used for weather and climate prediction.

*Code and data availability.*  The source codes for TSMP and the setups used for this study are freely available from https://www.terrsysmp.org/ with registration. The component models of TSMP has to be downloaded separately. The COSMO model is distributed to research institutions free of charge under an institutional license issued by the Consortium COSMO and administered by DWD. For more information see http://www.cosmo-model.org/content/consortium/licencing.html. The radar forward operator EMVORADO is based on source code derived from the COSMO model, hence redistribution is limited by the COSMO license. The ARTv3.1 model can be obtained from https://www.imk-tro.kit.edu/english/5224.php by writing an e-mail to bernhard.vogel@kit.edu. The CLM v3.5 model can be downloaded from https://www.cgd.ucar.edu/tss/clm/distribution/clm3.5/. The ParFlow model can be downloaded from https://parflow.org. The OASIS3-MCT coupler can be downloaded from https://oasis.cerfacs.fr/en/.

The COSMO license also includes access to lateral boundary data data provided by DWD. COSMO-DE EPS data used for the initial and lateral boundary conditions data for the COSMO model experiments in this study can be downloaded from the DWD database (https://www.dwd.de/DE/leistungen/pamore/pamore.html). The data used for soil-vegetation states are available https://doi.org/10.5880/TR32DB.40. The CAMS-REG v4.2 data can be downloaded from https://eccad3.sedoo.fr/. The MOZART-4 data is available from https://www.acom.ucar.edu/wrf-chem/mozart.shtml.

The Python package "emiproc" for emission pre-processing is available through C2SM Github Organization https://github.com/C2SM-RCM/cosmo-emission-processing. The COSMO model Processing Chain version 2.2 is available from https://github.com/C2SM/processing-chain. The source codes for pre-processing and analysis of model data, including scripts for plotting of figures are available from Github Organization https://github.com/prabshr/prom.

*Author contributions.*  PS conceptualized and designed the study, extended the TSMP modeling system with ART v3.1 and FO, conducted the model simulations and FO runs, carried out the analysis and wrote the manuscript. JM made adaptations to the FO and aided in model analysis and writing the manuscript. DB contributed in setup of TSMP runs with ART v3.1 module and aided in analysis of model results and writing the manuscript.

*Competing interests.*  The authors declare that they have no conflict of interest.



*Acknowledgements.* The research was carried out in the framework of the Priority Programme SPP-451 2115 "Polarimetric Radar Observations meet Atmospheric Modelling (PROM)" funded by the German Research Foundation (DFG). Prabhakar Shrestha acknowledges his support for PROM sub-project ILACPR (Grant SH 1326/1-1). J. Mendrok and carried out her work under PROM sub-project Operation Hydrometeors (Grants BL 945/2-1 ). We gratefully acknowledge the computing time (project HBN33) granted by the John von Neumann Institute for Computing (NIC) and provided on the supercomputer JUWELS at Jülich Supercomputing Centre (JSC). We would also like to thank Heike Vogel for her support with the use of ART v3.1 module in COSMO. We also thank the Björn Nillius and Birger Bohn for their effort in establishing and maintaining MAINZ and FJZ-JOYCE AERONET sites. The post-processing of model output data and input/output for FO was done using the NCAR Command language (Version 6.4.0). Dust data and/or images were provided by the WMO Barcelona Dust Regional Center and the partners of the Sand and Dust Storm Warning Advisory and Assessment System (SDS-WAS) for Northern Africa, the Middle East and Europe.





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



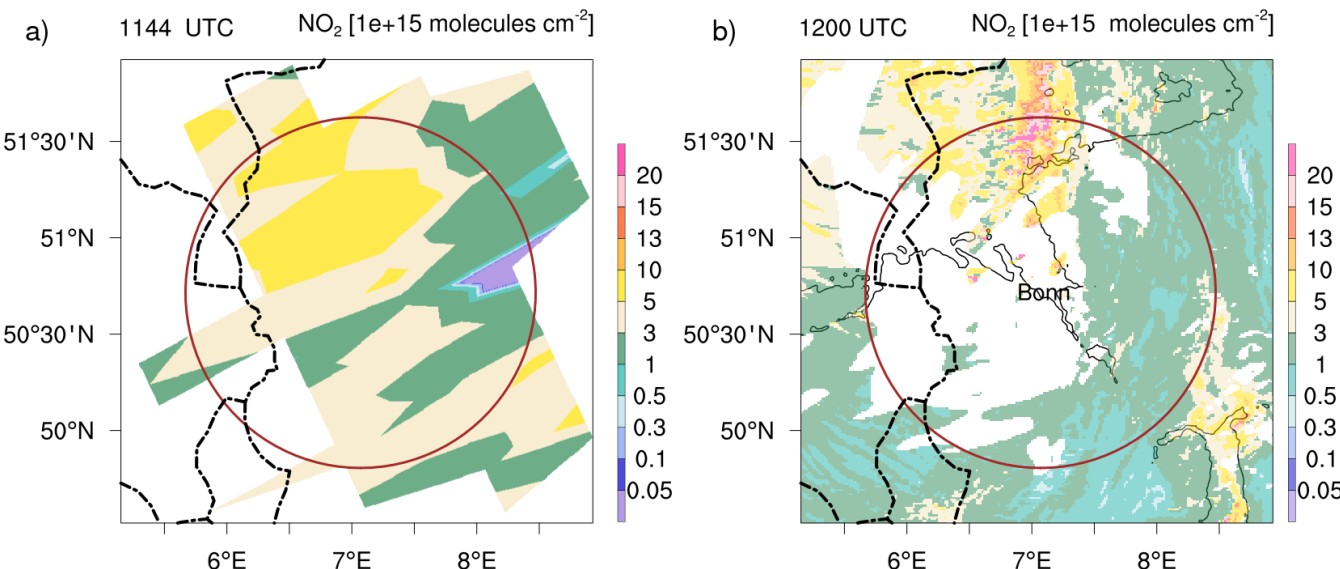

**Figure 1.** Satellite and model estimates of integrated vertical tropospheric column (VTC) for $NO_2$ over the Bonn Radar domain on 4 July 2015.

**Figure 2.** a-b) Satellite and model estimate of aerosol optical depth (AOD at 550 nm) over the Bonn Radar domain on 2015 4 July. The two available AERONET stations over the Bonn Radar domain is also shown. c-d) Time-series of measured and simulated ensemble AOD over FJZ-JOYCE and MAINZ AERONET station.



**Figure 3.** (a) Spatial pattern of aerosol number concentration for nuc./acc.(pure + mixed; $N_{na}$) at 2000 m height on 2015/07/05 1400 UTC. The square with x-mark at center indicates the sampling location east of BoxPol with the extent of 9x9 box. (b) Average aerosol size distribution of different modes and PM2.5 concentration for the 9x9 box. (c) Meridional cross-section of aerosol number concentration and PM2.5 concentration passing through the sampling location. (d) Ensemble vertical profile of aerosol number concentration for nuc./acc.(pure + mixed; $N_{na}$), soot ($N_{soot}$), dust ($N_{dust}$) and PM2.5 concentration at 'x'. (e) Ensemble time-series of aerosol number concentration for nuc./acc.(pure + mixed) at 2000 m height at BoxPol location. The blue and red line corresponds to time at 1400 and 1500 UTC respectively.



**Figure 4.** Same as in Fig. 4 but at 2015/07/05 1500 UTC



**Figure 5.** Spatial pattern of accumulated precipitation: a) Ensemble average from model; b) RADOLAN estimates. The black marker shows the location of BoxPol. c) Frequency distribution of the simulated and observed accumulated precipitation. The inset shows the domain average accumulated precipitation for each ensemble member (light grey color bar) and observation (black color bar) with one standard deviation (solid line above the bars).

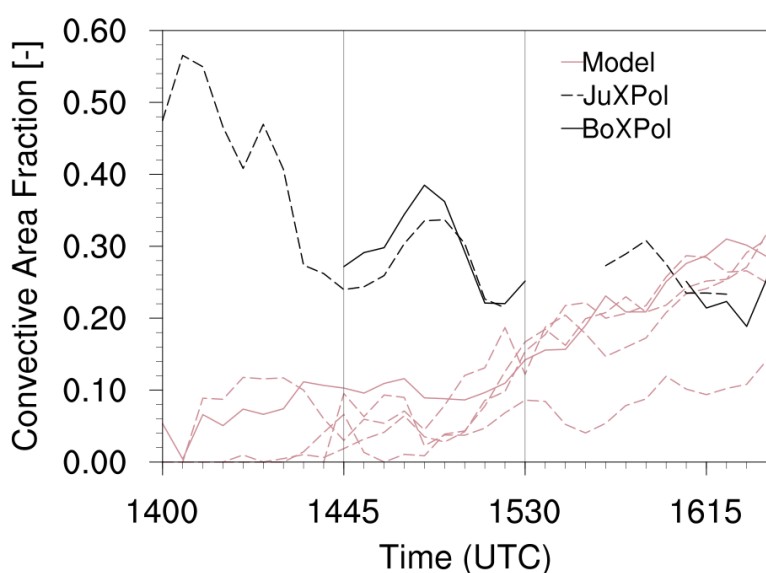

**Figure 6.** Convective Area Fraction (CAF) of model ensemble members and observations. The two vertical bars defines the time-period used to compute CFADs for observation and model. The ensemble member with solid line is used for polarimetric signature comparison. The CAF estimates from BoXPol or JuXPol are shown upon coverage and data availability. The gaps in the radar data represents times, when the polarimetric signatures are strongly attenuated or if the storm extent is only partially covered by the radar.



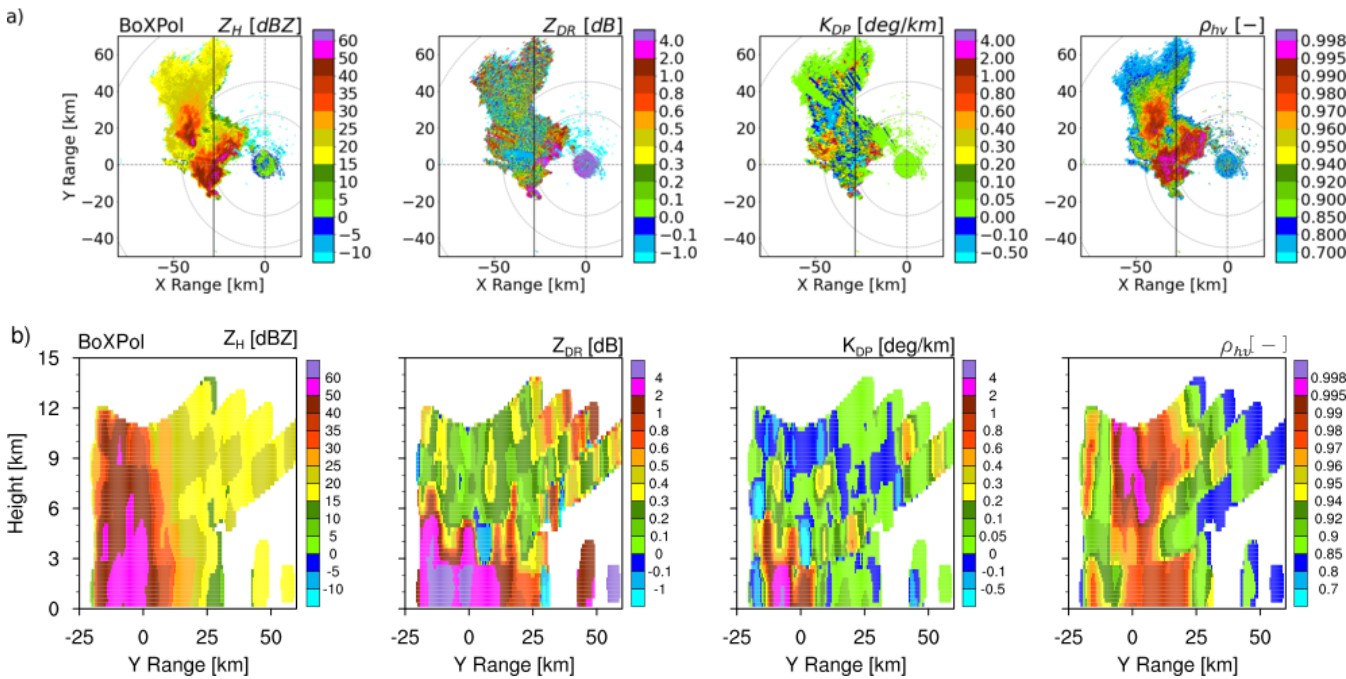

**Figure 7.** a) Plan position indicator (PPI) plots of horizontal reflectivity, differential reflectivity, sp. differential phase and cross-correlation coefficient at 8.2 degree elevation measured by BoXPol on 5 July 2015 at 1530 UTC. The dotted gray circles represent slant ranges for the chosen elevation angle, associated with heights of 1 km (lower level) , 4.5 km (melting level) and 7 km (upper level). b) Cross-section of the same polarimetric variables from the gridded data. The vertical solid black line along the Y Range in a) indicates the location of cross-section plots.



**Figure 8.** a,b) Model simulated horizontal reflectivity, differential reflectivity, sp. differential phase and cross-correlation coefficient at low level (1000 m a.g.l.) and near melting layer (4000 m a.g.l.) on 5 July 2015 at 1530 UTC. The 'x' mark refers to the BoXPol location. The gray solid line indicates the location of cross-section. c) Cross-section of the same polarimetric variables. d) Cross-section of model simulated hydrometeor density [QR(rain), QS (snow), QG (graupel) and QH (hail)]. Also shown are the $0°C$ line (solid black line) indicating the melting layer, contours of vertical velocity [5,10,20,40 m/s] with QS and contours of cloud ice density (QI) with QH.



**Figure 9.** Contoured frequency altitude diagrams (CFADs) of horizontal reflectivity, differential reflectivity, sp. differential phase and cross-correlation coefficient from 1445 to 1530 UTC on 5 July 2015. CFADs from the model are shown for 5 ensemble members.

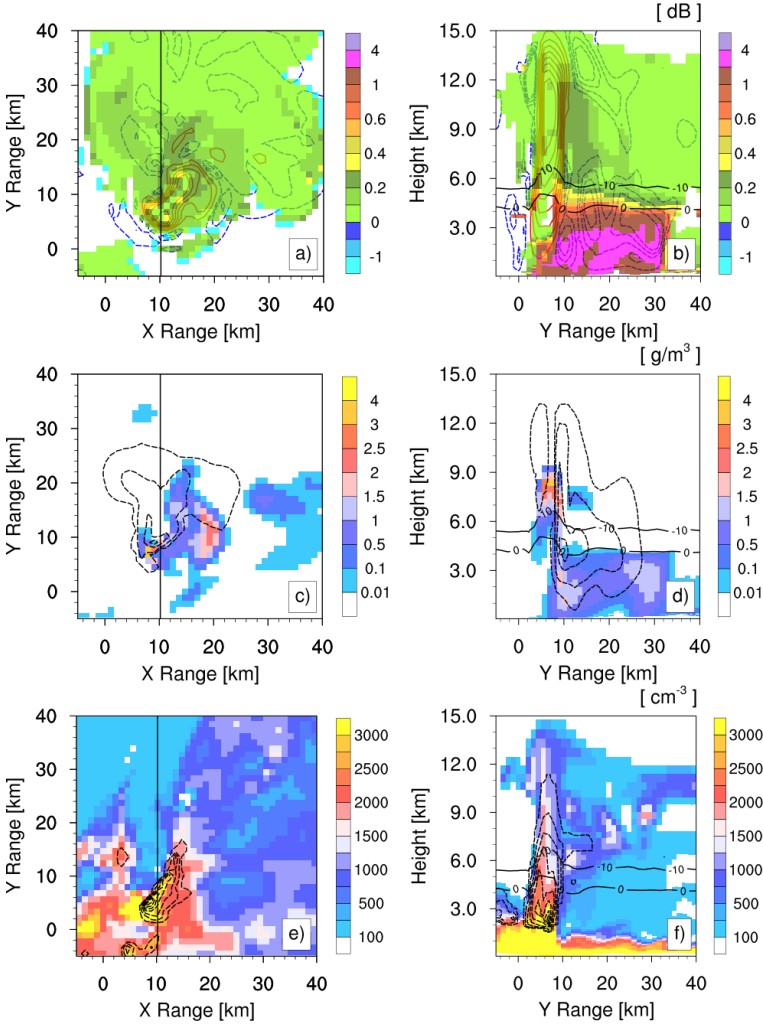

**Figure 10.** Plan view (left panel) and vertical cross-section (right panel) of aerosols, model states and polarimetric variables. The plan views are shown at 6 km height and all cross-sections are passing through the solid line shown in the plan view. The $0\,^\circ C$ and $-10\,^\circ C$ isotherm is also shown in all cross-sections. a,b) Differential reflectivity and vertical velocity. The contoured solid/dashed red/blue lines indicate updraft and downdraft respectively. The vertical wind speed contours are shown at the following intervals (-7.,-5.,-3.,-1.,5.,10.,15.,20.,25.,30.,35.,40.) in m/s. c,d) Rain and Hail mixing ratios in filled and solid line contours respectively. e,f) Aerosol and cloud number concentrations in filled and dashed line contours respectively. The cloud number concentration is contoured at interval of $500\,cm^{-3}$ with minimum of $100\,cm^{-3}$ The aerosol concentration is shown for the pure and mixed nucleation and accumulation model aerosols.



**Figure 11.** a) Modified gamma particle size distribution as a function of particle diameter ($D_p$) for the default and narrow cloud drop size distribution (CDSD). The bulk number concentration and mass density is $300\,cm^{-3}$ and $1\,gm^{-3}$ respectively. b) Spatial pattern of ensemble averaged accumulated precipitation for the default and narrow CDSD (solid contour lines with intervals at 20 , 30 and 35 mm); c) Contoured frequency altitude diagrams (CFADs) of horizontal reflectivity, differential reflectivity, sp. differential phase and cross-correlation coefficient from 1445 to 1530 UTC on 5 July 2015. CFADs are shown for 5 ensemble members for narrow CDSD.





**Table 1.** Parameters of the size-mass and velocity-mass relationships following Eqs. (2) and (3) used in the SB2M. These refer to $D$ in units of m, $x$ in kg, and $v_T$ in $\mathrm{m\,s^{-1}}$. The last two columns contain the shape parameters of the assumed mass distribution. $D_{x,min} = a_g x_{min}^{b_g}$ and $D_{x,max} = a_g x_{max}^{b_g}$ are the diameters corresponding to the mass limits $x_{min}, x_{max}$ and are added for better interpretation.

| | $\mathbf{a_{geo}}$ | $\mathbf{b_{geo}}$ | $\mathbf{a_v}$ | $\mathbf{b_v}$ | $\mathbf{x_{min}}$ | $\mathbf{x_{max}}$ | $\mathbf{D_{x,min}}$ | $\mathbf{D_{x,max}}$ | $\mu$ | $\nu$ |
|---|---|---|---|---|---|---|---|---|---|---|
| **Cloud Liquid** | 0.124 | 1/3 | $3.75 \cdot 10^5$ | 2/3 | $4.2 \cdot 10^{-15}$ | $2.6 \cdot 10^{-10}$ | $2.0 \cdot 10^{-6}$ | $8.0 \cdot 10^{-5}$ | 0 | 1/3 |
| **Rain** | 0.124 | 1/3 | 114.0 | 0.234 | $2.6 \cdot 10^{-10}$ | $3.0 \cdot 10^{-6}$ | $8.0 \cdot 10^{-5}$ | $1.8 \cdot 10^{-3}$ | 0 | 1/3 |
| **Cloud Ice** | 0.835 | 0.390 | 27.7 | 0.216 | $1.0 \cdot 10^{-12}$ | $1.0 \cdot 10^{-6}$ | $1.7 \cdot 10^{-5}$ | $1.6 \cdot 10^{-3}$ | 0 | 1/3 |
| **Snow** | 2.4 | 0.455 | 4.2 | 0.092 | $1.0 \cdot 10^{-10}$ | $2.0 \cdot 10^{-5}$ | $6.8 \cdot 10^{-5}$ | $1.8 \cdot 10^{-2}$ | 0 | 1/2 |
| **Graupel** | 0.142 | 0.314 | 86.89 | 0.268 | $1.0 \cdot 10^{-9}$ | $5.0 \cdot 10^{-4}$ | $2.1 \cdot 10^{-4}$ | $1.3 \cdot 10^{-2}$ | 1 | 1/3 |
| **Hail** | 0.1366 | 1/3 | 39.3 | 1/6 | $2.6 \cdot 10^{-9}$ | $5.0 \cdot 10^{-4}$ | $1.9 \cdot 10^{-4}$ | $1.1 \cdot 10^{-2}$ | 1 | 1/3 |





**Table 2.** Overview of EMVORADO melting scheme setup used in this study.

|  |  | cloud ice | snow | graupel | hail |
|---|---|---|---|---|---|
| $T_{\text{meltbegin}}$ | [°C] | 0.0 | 0.0 | -10.0 | -10.0 |
| $T_{\text{meltdegTmin}}$ | [−] | 0.0 | 0.0 | 0.2 | 0.2 |
| $\min(T_{\max})$ | [°C] | 2.0 | 3.0 | 3.0 | 5.0 |
| $\max(T_{\max})$ | [°C] | 5.0 | 10.0 | 15.0 | 30.0 |