# Peer review of "Aerosol characteristics and polarimetric signatures for a deep convective storm over north-western part of Europe – modeling and observations"

_Atmospheric Chemistry and Physics, 2022_

## Author Comment (AC1)

The comments by the reviewer are in black and our replies to the reviewers are in **bold blue**.

**RC1**
The manuscript, "Aerosol characteristics and polarimetric signatures for a deep convective storm over north-western part of Europe – modeling and observations," details the development and evaluation of an extended modeling system. This enables high-resolution modeling of aerosol-cloud interactions in the Terrestrial Systems Modeling Platform (TSMP), after extension with a chemical transport model and the polarimetric radar forward operator. The TSMP was evaluated at convection-permitting horizonal resolution against observations of a thunderstorm which took place in July 2015 over northwestern Germany. Overall, the extension of the model with the chemical transport model and polarimetric radar forward operator adds valuable capability to model predictions.

**We are very thankful for the reviewer's critique.**

Beyond this, however, this reviewer found it difficult to pull out what the key conclusions of this study actually were, and what benefits the extended TSMP platform provided. The manuscript suffers from too much unnecessary detail, and sometimes repeated itself, and not enough necessary detail. Not only did the lack of clarity and explanation make it difficult to identify the key findings of the study, it also made it difficult to evaluate the claims and evidence that were more clearly presented.

**In the revised manuscript, we have highlighted the key findings from the study: 1) Convective storm event in the model generates "aerosol tower" like features; 2) Model is able to capture the spatial pattern of precipitation and polarimetric features but with some biases; 3) Constraining the cloud droplet size distribution (CDSD) did not produce any drastic difference in the precipitation and CFADs of polarimetric variables but produced improvement in the simulated ZDR column-like features. Thus, running simulations with prognostic aerosol and use of forward operator can also additionally help to constrain the cloud droplet size distribution in the model.**

**The methods and results sections have been better explained, and extended details in the study have been moved to the supplementary to simplify the manuscript and better emphasize the main messages.**

For example, none of the polarimetry aspects of the study were clearly explained; prior knowledge of both polarimetric radar and how they are interfaced to/emulated by models seems to be necessary. Even more specifically as an example, polarimetric radar measurements such as horizontal reflectivity, differential reflectivity, specific differential phase, and the cross-correlation coefficient are introduced and the TSMP's performance evaluated for these parameters, but the manuscript does not explain to which meteorological or chemical parameters they relate, what these parameters tell us about the atmosphere and the model's performance. This reviewer is familiar with regional and high-resolution modeling and aerosol-chemistry-convection interactions but not polarimetric radar, and assumes that this will be the

case for at least some readers, so that much of the manuscript dealing with the polarimetric model implementation (even how it's interfaced with the model is unclear) and the related results are a bit shrouded in mystery. As a somewhat less critical example, the manuscript does not define what it considers to be northwestern Germany for the purpose of the study, and later on adds in mention of two additional German cities/radar sites, with insufficient location information for the deep convective event itself; a map may have been useful. Methods and results are not clearly explained – how are the column NO2 observations from satellite swaths and AERONET point observations compared to a model grid box, as another example. Why was this study undertaken in the first place is also left unclear (though such a modeling platform does have great potential).

**We agree with the reviewer that lack of prior knowledge of polarimetric radar and interfacing of FO with the model might hinder some readers. While polarimetric radar observations are still comparably new, we think that a detailed explanation and further their sensitivities is beyond the scope of this manuscript. So, to address the reviewer's concern, we have added references to literature on radar polarimetry in general (e.g. Ryzhkov and Zrnic 2019; Kumjian et al. 2013; Trömel et al. 2021) and its use for microphysical fingerprinting in particular. We have also added a discussion about the specific polarimetric signature like the ZDR column that plays a role in our study. Further, we have added additional details on the forward operator (FO), including references to papers with in-detail descriptions of the FO and e.g. its coupling to the COSMO model (Zeng et al. 2016; Trömel et al. 2021; Shrestha et al. 2022a,b), to allow the reader to better understand this part without (further) (over)loading this manuscript. We have also now better explained the benefits of the extended TSMP platform.**

**The choice of study region has been better motivated in the revised manuscript. Fig. R1.1 shows the model domain with the extents of the two polarimetric radars: JuXPol and BoXPol. The dotted lines show the extent of the model domain used for evaluation purposes excluding the relaxation zone.**

[Figure]

**Figure R1.1: Bonn Radar Domain (included in the revised manuscript).**

**In the manuscript, column NO₂ observations from satellite swaths are compared with the model qualitatively (more quantitative comparison is discussed below) and AERONET point observations are compared to a model grid box nearest to the AERONET location.**

**Finally, the study was undertaken to improve our understanding of aerosol-cloud-precipitation interactions, by overcoming shortcomings in the representation of aerosol dynamics in the model and by evaluating the model in polarimetric radar space.**

These examples are meant to be illustrative, as this problem is ubiquitous in the manuscript, and greatly impedes identification, interpretation, and evaluation of results.

**We are thankful to the reviewer's constructive criticism. We have made an effort to bring overall clarity in the manuscript in terms of methods, results and discussion of findings.**

Two Specific Comments:

Line 111: It is not really accurate to say that AERONET is considered to have a better accuracy than MODIS – one instrument is essentially a point-based observation and so is often likely more accurate for that specific location than a satellite, while the other observes over a larger swath and so loses some of the horizontal detail but can fill in the gaps between AERONET instruments. And both require a retrieval to turn the raw observations into useful measurements of aerosol properties, which introduces its own uncertainties. Which instrument is "better" for a study depends on what you're trying to compare or investigate. It's more accurate to say that MODIS and AERONET are complementary, and each has its advantages and disadvantages.

**We agree with the reviewer that MODIS and AERONET are complementary to each other, and provide valuable data to compare spatial patterns and time-series individual locations, respectively. However, we maintain our statement that AERONET AOD observations are more accurate than MODIS. This statement has been made in many previous studies, see Giles et al. (2019) and references therein. It is much easier to retrieve AOD by looking from below the atmosphere to the sun (as done by AERONET) than observing reflected sun-light (as done by MODIS), where it is difficult to disentangle surface reflectance from aerosol scattering effects. AERONET has therefore always served as a ground-truth for MODIS observations. AERONET AOD observations have a 1-$\sigma$ uncertainty of 0.02 (Giles et al., 2019), whereas MODIS has an uncertainty of 0.05$\pm$15 % (Levy et al., 2013), and data quality gradually decreases with increasing surface reflectance (Wei et al., 2018). Note also that by measuring sky radiance at multiple scattering angles in addition to direct-sun observations, AERONET can much more accurately determine further aerosol properties such as aerosol size distribution and phase-function than MODIS, where aerosol types and corresponding properties are prescribed in look-up tables as a function of time and space (Levy et al. 2007, Levy et al. 2013). We changed the sentence in revised manuscript to "These measurements have a better accuracy than MODIS (Giles et al. 2019) but are only available at a few locations."**

Line 245: Comparison of satellite-based column observations to the NO2 columns computed from the model output is more than simply a first-order evaluation, but this again depends to an extent on what exactly you want to evaluate and be able to say with your study. Column quantities are quite useful for many applications, and again, the satellite-based column observations fill in gaps between ground-based or other types of in situ instruments and can, for example, provide information on chemical transport and transformation. The brief evaluation of the NO2 column comparisons already demonstrate a number of potential results and areas for model refinement that could be classified as more than first-order. Also, why were other trace gases not evaluated, such as O3 or HCHO which are available from satellite data products? And equally importantly, the satellite products come with specified uncertainties and data flags, so it is not enough to only acknowledge the uncertainty in the satellite estimates – the uncertainty in the comparison can therefore be quantified and would facilitate evaluation of the comparison.

**The reviewer is right that this is more than just a first-order comparison since we are directly comparing model-simulated NO₂ VTCs with satellite retrieved VTCs over the same domain and at approximately the same time. What we tried to say is that this is a very limited evaluation since we are comparing the model with observations for a single day only. The product indeed comes with an estimated uncertainty for each VTC that could be used for the comparison between satellite and model. According to Boersma et al. (2011), Boersma et al. (2018) and Lamsal et al. (2021) typical uncertainties are of the order of 30% under clear-sky-conditions. This should also hold for the data used in our study, since we followed the recommenations in the Readme document available at [https://disc.gsfc.nasa.gov/datasets/OMNO2_003/summary](https://disc.gsfc.nasa.gov/datasets/OMNO2_003/summary)(i.e. filtering for data with a quality flag of zero and with a cloud radiance fraction < 0.5, as already mentioned in the manuscript). In order to compare simulated and observed NO2 VTCs more quantitatively, we have additionally created a scatter plot between simulated and observed VTCs (see Fig. R1.2). For this, we have interpolated the model output over the individual OMI pixels using an inverse distance squared algorithm.**

[Figure]

**Figure R1.2: Scatter plot with regression line of NO2 VTCs between satellite and model estimates (r= 0.46). The model output was interpolated over the individual OMI pixels using an inverse distance squared algorithm.**

We haven't included OMI HCHO and ozone for the following reasons: The HCHO product is extremely noisy. The retrieval uncertainty is 50-105%, with the lower end being valid only for highly polluted locations. HCHO products are therefore usually only presented as monthly, seasonal or yearly averages (e.g. De Smedt et al. 2021). An illustration of the extremely high noise in OMI data compared to MAX-DOAS measurements is shown in their Fig. 15. The same argument holds for $SO_2$. Comparing ozone would be quite interesting, but there is no official OMI tropospheric ozone product. Comparing the total column $O_3$ from OMI would not be meaningful as the column is strongly dominated by the stratosphere. Note also that due to the long lifetime of ozone, we would expect only very small gradients in the model domain.

The above discussion has been included in the revised manuscript.

References:

Boersma, K. F., Eskes, H. J., Dirksen, R. J., van der A, R. J., Veefkind, J. P., Stammes, P., Huijnen, V., Kleipool, Q. L., Sneep, M., Claas, J., Leitão, J., Richter, A., Zhou, Y., and Brunner, D.: An improved tropospheric NO2 column retrieval algorithm for the Ozone Monitoring Instrument, Atmos. Meas. Tech., 4, 1905–1928, https://doi.org/10.5194/amt-4-1905-2011, 2011

Boersma, K. F., Eskes, H. J., Richter, A., De Smedt, I., Lorente, A., Beirle, S., van Geffen, J. H. G. M., Zara, M., Peters, E., Van Roozendael, M., Wagner, T., Maasakkers, J. D., van der A, R. J., Nightingale, J., De Rudder, A., Irie, H., Pinardi, G., Lambert, J.-C., and Compernolle, S. C.: Improving algorithms and uncertainty estimates for satellite NO2 retrievals: results from the quality assurance for the essential climate variables (QA4ECV) project, Atmos. Meas. Tech., 11, 6651–6678, https://doi.org/10.5194/amt-11-6651-2018, 2018.

De Smedt, I., Pinardi, G., Vigouroux, C., Compernolle, S., Bais, A., Benavent, N., Boersma, F., Chan, K.-L., Donner, S., Eichmann, K.-U., Hedelt, P., Hendrick, F., Irie, H., Kumar, V., Lambert, J.-C., Langerock, B., Lerot, C., Liu, C., Loyola, D., Piters, A., Richter, A., Rivera Cárdenas, C., Romahn, F., Ryan, R. G., Sinha, V., Theys, N., Vlietinck, J., Wagner, T., Wang, T., Yu, H., and Van Roozendael, M.: Comparative assessment of TROPOMI and OMI formaldehyde observations and validation against MAX-DOAS network column measurements, Atmos. Chem. Phys., 21, 12561–12593, https://doi.org/10.5194/acp-21-12561-2021, 2021.

Kumjian, M. R. (2013). Principles and Applications of Dual-Polarization Weather Radar. Part I: Description of the Polarimetric Radar Variables. Journal of Operational Meteorology, 1.

Levy, R. C., Remer, L. A., and Dubovik, O. (2007), Global aerosol optical properties and application to Moderate Resolution Imaging Spectroradiometer aerosol retrieval over land, J. Geophys. Res., 112, D13210, doi:10.1029/2006JD007815.

Levy, R. C., Mattoo, S., Munchak, L. A., Remer, L. A., Sayer, A. M., Patadia, F., and Hsu, N. C.: The Collection 6 MODIS aerosol products over land and ocean, Atmos. Meas. Tech., 6, 2989–3034, https://doi.org/10.5194/amt-6-2989-2013, 2013.

Ryzhkov, A. V., & Zrnic, D. S. (2019). Radar polarimetry for weather observations (Vol. 486). Cham, Switzerland: Springer International Publishing.
Shrestha, P., Trömel, S., Evaristo, R., & Simmer, C. (2022). Evaluation of modelled summertime convective storms using polarimetric radar observations. Atmospheric Chemistry and Physics, 22(11), 7593-7618.

Shrestha, P., Mendrok, J., Pejcic, V., Trömel, S., Blahak, U., & Carlin, J. T. (2022). Evaluation of the COSMO model (v5. 1) in polarimetric radar space–impact of uncertainties in model microphysics, retrievals and forward operators. Geoscientific Model Development, 15(1), 291-313.

Trömel, S., Simmer, C., Blahak, U., Blanke, A., Doktorowski, S., Ewald, F., ... & Quaas, J. (2021). Overview: Fusion of radar polarimetry and numerical atmospheric modelling towards an improved understanding of cloud and precipitation processes. Atmospheric Chemistry and Physics, 21(23), 17291-17314.

Wei, J., Sun, L., Peng, Y., Wang, L., Zhang, Z., Bilal, M., & Ma, Y. (2018). An improved high-spatial-resolution aerosol retrieval algorithm for MODIS images over land. Journal of Geophysical Research: Atmospheres, 123, 12,291– 12,307. https://doi.org/10.1029/2017JD027795

Zeng, Y., Blahak, U., & Jerger, D. (2016). An efficient modular volume-scanning radar forward operator for NWP models: description and coupling to the COSMO model. Quarterly Journal of the Royal Meteorological Society, 142(701), 3234-3256.

---

## Author Comment (AC2)

The comments by the reviewer are in black and our replies to the reviewers are in **bold blue**.

**RC2**

Review of Shrestha et al,

In this study a convective situation is simulated with an ensemble of TSMP-ART runs over the Bonn region of Germany. The simulations are validated using the Bonn Polarimetric radar, aeronet and modis observations. In general, the ensemble runs are able to simulate the convective cell structure reasonably well, while the aerosol concentrations and precipitation are underestimated. When the model runs are run through a Forward Operator, difference between the simulated and observed cloud structure are presented. The paper is extremely well written but the story line and figures are very complex. I recommend the article for publication in ACP once the comments below are addressed.

**We are thankful for the reviewer's comments. Below, we address the reviewer's specific comments.**

General comments:

The paper is quite technical with lots of details (e.g. discussion about specific features in the plots), which often distract from the overall research questions and ending conclusions.

**We are thankful for the reviewer's critique, we agree that this interdisciplinary research work involving aerosol dynamics and radar polarimetry might have led to a complex story line. In the revised manuscript, we have made an effort to bring clarity and simplicity to the story line and discussion. In addition, extended technical details have been moved to supplementary.**

The main research questions are not really answered and it comes off more of as, "we have these observations, let's see how our more complex model is doing" rather than geared to address a particular research question e.g. does using prognostic aerosols improve the reproducibility of the storm, does constraining the cloud droplet distribution improve storm signatures, does a Forward Operator allow us to understand model deficiencies more easily? Especially as research question 2) is never really answered, as it is not clear if the "capabilities" lead to an improvement to simulations without these additions. There is no doubt that getting these components of the model to run together is a complex and cumbersome process, but without showing that this is an improvement or how the Forward Operator polarimetric variables improve the assessment of aerosol-cloud-precipitation interactions, the paper is lacking a clear direction.

**We are thankful for the reviewer's critique. In this study, we extended the state-of-the-art terrestrial systems modeling platform with a chemistry transport module and a polarimetric forward operator. The model was then used to evaluate synthetic polarimetric signatures of a deep convective storm event over Germany with observations from X-band radar to better understand aerosol-cloud-precipitation interaction. The simulated precipitation and CFADs of**

**polarimetric variables do have a similar resemblance to simulations using fixed continental aerosols (Shrestha et al. 2021). So, using a prognostic aerosol model does not necessarily show an improvement in the reproducibility of the storm. However, it does show that the assumed aerosol homogeneous aerosol physical and chemical properties during a convective storm event is not realistic. Additionally, constraining the cloud droplet size distribution (CDSD) also did not produce any drastic difference in the precipitation and CFADs of polarimetric variables. However, the narrow CDSD does show improvement in the simulated $Z_{DR}$ column-like features, which is more well defined than the default experiment with larger mean rain drop size (0.5 ~ 1 mm) above the melting layer (Please see further discussion below). Thus, the Forward Operator helps us to recognize such model deficiencies (e.g., the missing feedback between aerosols and CDSD).**

 One of the main findings from the aerosol module is that the convective updrafts produce aerosol towers and that the properties of the aerosols change after cloud processing. However, these themes are not carried through the manuscript. There is also very little discussion on how the aerosol cloud processing is parameterized or compared to previous studies. Also, it is not immediately clear if these aerosols will remain interstitial when exposed to such high updrafts. If they are activated as cloud droplets/nucleated into ice crystals, will they be scavenged and removed?

**As presented in Section 2.2, the cloud nucleation parameterization is based on the works of Fountoukis and Nenes (2005), Barahona and Nenes (2007), Kumar et al. (2009) and Barahona et al. (2010). Similarly, the ice nucleation parameterization is based on Barahona and Nenes (2009). The detailed implementation of the above schemes and aerosol cloud processing is discussed in detail in Bangert et al. (2012). And, yes, the activated aerosols as cloud droplets/nucleated into ice crystals will be scavenged and removed. Besides the environmental and microphysical factors, the aerosol activation would also depend on its physical and chemical properties, which varies with elevation in the model. This could be partly contributing to variable partitioning between the interstitial and activated aerosols as cloud droplets.**

**We have added the following additional description in the revised manuscript:**
**"The comprehensive activation parameterization works for a parcel of air containing an external mixture of soluble and insoluble aerosols. The activation rate is applied directly for newly formed clouds, while for existing clouds, the activation rate at cloud base is calculated based on advection and turbulent diffusion of particles into cloud base (Bangert et al. 2012). Further, for strong updrafts, in-cloud activation is also computed, for which growth of existing cloud droplets is considered by assuming they act as giant CCN that deplete supersaturation (Bangert et al. 2012). The activated aerosols as cloud droplets/nucleated into ice crystals are scavenged and removed. The washout of the aerosols by precipitation was also turned on for the simulation in this study. Besides the environmental and microphysical factors, the aerosol activation would also depend on its physical and chemical properties, which varies with elevation in the model. This can contribute to variable partitioning between the interstitial and activated aerosols as cloud droplets."**

Are the polarimetric variables actually helping point to why the model is not matching the observations? There is a lot of discussion on how these variables differ but not to what questions they can answer. At the same time, as the Forward Operator introduces many uncertainties in itself as mentioned in the discussion, is it worth including it as a diagnostic tool? Consider building on the advantages of including this in the modelling process/ post processing.

**Polarimetric measurements contain (rich) information about microphysical processes, e.g. about aggregation, riming, and melting (see e.g. Kumjian (2013a,b). FOs provide observation equivalents of the model states (which in turn contain imprints of the modeled microphysical processes), enabling comparison of microphysical process signatures in both radar simulations and observations in 3D space and time. Using model states of hydrometeors and polarimetric features, it also helps us to understand why the model is not matching with the observations. Eg. the $Z_{DR}$ column is one of the dominant polarimetric features in the observations, which is captured in the model but with different width, extent and intensity. The weaker $Z_{DR}$ column could be partly attributed to the small size of raindrops above the melting layer, which appears to be improved while using narrow cloud drop size distribution in the sensitivity study.**

At the same time, one of the main hypothesis for the discrepancy between observed and simulated polarimetric variables is the cloud droplet size distribution. Even though this explicitly tested, the discussion on this topic is not well connected. Even when running this sensitivity study, the polarimetric variables do not improve significantly. Does this really mean it is necessary to run simulations with prognostic aerosol as stated in the conclusion? Rather is it better to further adjust the cloud droplet size distribution?

**Constraining the cloud droplet size distribution (CDSD) did not produce any drastic difference in the precipitation and CFADs of polarimetric variables. However, the narrow CDSD does show improvement in the simulated $Z_{DR}$ column-like features, which is more well defined than the default experiment with larger mean rain drop size (0.5 ~ 1 mm) above the melting layer.**

**Fig. R2.1 below shows the plan and cross-section view of differential reflectivity at 1525 UTC for the narrow CDSD experiment. The plan view is shown at 6 km height. The 0 C and −10 C isotherm is also shown in the cross-section. The contoured solid/dashed red/blue lines**

**indicate updraft and downdraft respectively.**

[Figure]

**Figure R2.1: Plan and cross-section view of differential reflectivity at 1525 UTC for the narrow CDSD experiment (included in the revised manuscript)**

**Thus, running simulations with prognostic aerosol can also additionally help to constrain the cloud droplet size distribution.**

Many of the figures are poorly labeled and described in the captions as noted below. This makes the discussion difficult to follow.

**We have improved the captions and the labels of the figures in the revised manuscript.**

Minor comments:

Line 60: remove "s" from lightning.
**Corrected.**

Line 63: would be nice to have a domain of the modelling in the manuscript, rather than just a reference to a previous study.

**Figure of the domain has been added in the revised manuscript (see Fig. R2.2).**

[Figure]

**Figure R2.2: Bonn Radar Domain (included in the revised manuscript).**

Line 137-149: As this study looks at the aerosol transport in convective updrafts, the description on aerosol activation (both CCN and INP), subsequent removal via precipitation and aerosol scavenging should be described. There is no doubt that updrafts help to lift boundary layer aerosol into the free troposphere but the fraction of these aerosol, especially large ones that do not act as CCN/INP should be discussed.

**See comments above.**

Line 163: "the" -> "that"

**Corrected.**

Line 179: remove "this"

**Corrected.**

Figure 1: please add a and b in the caption

**Added.**

Figure 3: In panel (a) add units for number conc. Panel (c), is that the 0 degree isotherm? Why is the center latitude for the cross section not at the center of the box? Legend for altitude lines are very hard to see. Also, is crust meant to be dust? Make it clearer that the same colored lines are from ensemble runs, especially in the PM2.5 vertical distribution in panel (d).

**The unit is present in Panel (a). Yes, it is the 0 degree isotherm in Panel (c). It is also now explicitly mentioned in the caption of the revised manuscript. The "x" mark is not at the center, we have fixed it in the caption. Assuming that the reviewer meant the legend in Panel (b), we have enlarged the fonts in the revised manuscript. Yes, the crust here refers to dust -**

**we have changed it to dust in the revised manuscript. The same colored lines are from ensemble runs in Panel (d), this has now been explicitly mentioned in the revised manuscript.**

Line 267: Is the storm/ air flow moving to the north east? So the wind would be southwesterly or in a northeasterly direction? If this is not the case, would be nice to have sort of meteorological overview e.g. wind barbs to understand the direction of the flow.

**The storm is moving towards north east, with southwesterly winds, this has been corrected in the revised manuscript.**

Line 268: Why However? Are the updrafts not also helping to distribute the aerosol. Additionally, it looks like the role of updrafts is far more important at heights above 2km. That's probably not important, but it begs the questions as to why the 2 km height is chosen for this analysis (later in the text the 6 km level is discussed).

**Here, we wanted to emphasize that the change in aerosol pattern is not only affected by advection but also by propagating updrafts which transport the boundary layer aerosol above, as shown in the 2 km height patterns. We have rephrased the sentence in the revised manuscript. The 2 km height was chosen, because it represents the spatial pattern of aerosol near the boundary layer top height. Later, the 6 km height is used for associated discussion with polarimetric features at that level.**

Figure 4: the legend should reference Figure 3 and see comments for Fig. 3. CRUST seems cutoff in panel (b).

**Fixed.**

Line 269: "to determined" add "be"

**Added.**

Line 287: Please define "ABL", is this aerosol boundary layer?

**It's atmospheric boundary layer, now defined in the revised manuscript.**

Line 288-290: But on the previous day, the peak occurred earlier and lasted well into the early morning hours. It is not evident that this statement is supported from this two-day simulation period. At the same time, it is not really clear why this is important.

**Here, we want to emphasize that the diurnal cycle of aerosols is contributed by the ABL evolution and propagating convective updrafts. The sentence has been rephrased accordingly.**

Line 295: should it be PM2.5 mass (concentrations) or is PM2.5 concentration always reported in mass?

**It is corrected to PM2.5 mass concentrations in the revised manuscript.**

Figure 10: panels a and b the colorfill units are not immediately clear.  Why are there so few aerosols in the vertical cross section at ranges less than 10 km in panel f at the 6 km height while in panel e, the concentration at 6 km is close to 2000 cm-3?

**Panels a) and b) colorfill represents differential reflectivity. This has now been described more clearly in the revised manuscript. Panel e) was mistakenly shown for 2 km height, this has been now corrected in the revised manuscript.**

Figure 11: Please label the red gamma distribution as the CDSD. Also, it is hard to easily compare the CFAD ensembles with the CDSD and lognormal droplet size. Consider making a difference plot of the CFADS.

**Both distributions are CDSD in Panel (a). We have replaced Panel (c ) with CFADs of both experiments (see Fig. R2.3). The CFADs of the default experiment are shown in contoured lines only.**

[Figure]

**Figure R2.3: CFADs of polarimetric variables for narrow CDSD and default experiment (included in the revised manuscript)**

Line 441-442: it would be nice to have a discussion about how changing the cloud droplet size distribution assumptions had on this here.

[Figure]

**Figure R2.4: CAF time-series with the narrow cloud droplet size distribution.**

**Fig. R2.4 shows the CAF time-series with the new cloud droplet size distribution. Since the experiment is only presented later on, the following text has been added after Ln 470 where the CAF results were discussed earlier:**
**"The change in cloud droplet size distribution led to delay in the onset of CAF evolution, with some ensemble members exhibiting relatively higher CAF. However, the CAF time series exhibits different variability for each ensemble member, suggesting strong influence of lateral boundary conditions."**

Line 445: second Poll ref is missing a year.
**Added.**

Line 458-460: Again here, did things improve when a different cloud droplet size distribution was assigned? This is touched on later in the section but the discussion could be shortened to combine the influence of the CDSD runs.

**In general, the narrow CDSD does show improvement in the simulated ZDR column-like features, which is more well defined than the default experiment, as discussed above.**
**The following text has been added in Ln 473:**
**"The narrow CDSD does show improvement in the simulated ZDR column-like features, which is more well defined than the default experiment with larger mean rain drop size (0.5 ~ 1 mm) above the melting layer. "**

Line 484-486: There is no doubt that the strong updrafts help to loft aerosols to higher levels. However, with such strong updrafts (e.g. > 10 m/s), is it realistic that the aerosol are still interstitial and not activated as cloud droplets?

**Besides the environmental and microphysical factors, the aerosol activation would also depend on its physical and chemical properties, which varies with elevation in the model. This could be partly contributing to variable partitioning between the interstitial and activated aerosols as cloud droplets.**

Line 492: Consider rephrasing to: "…can also not be neglected"
**Corrected.**

**References:**

**Kumjian, M. R. (2013a). Principles and Applications of Dual-Polarization Weather Radar. Part I: Description of the Polarimetric Radar Variables. Journal of Operational Meteorology, 1(19):226-242, doi: 10.15191/nwajom.2013.0119**

**Kumjian, M. R. (2013b). Principles and Applications of Dual-Polarization Weather Radar. Part II: Warm- and Cold-Season Applications. Journal of Operational Meteorology, 1(20):243-264, doi: 10.15191/nwajom.2013.0120**